# Vaccinating Welders against Pneumococcus: Evidence from a Systematic Review and Meta-Analysis

**DOI:** 10.3390/vaccines11091495

**Published:** 2023-09-15

**Authors:** Matteo Riccò, Pietro Ferraro, Salvatore Zaffina, Vincenzo Camisa, Federico Marchesi, Davide Gori

**Affiliations:** 1Servizio di Prevenzione e Sicurezza Negli Ambienti di Lavoro (SPSAL), AUSL-IRCCS di Reggio Emilia, Via Amendola n.2, I-42122 Reggio Emilia, Italy; 2Occupational Medicine Unit, Direzione Sanità, Italian Railways’ Infrastructure Division, RFI SpA, I-00161 Rome, Italy; dott.pietro.ferraro@gmail.com; 3Occupational Medicine Unit, Bambino Gesù Children’s Hospital IRCCS, I-00152 Rome, Italy; salvatore.zaffina@opbg.net (S.Z.); vincenzo.camisa@opbg.net (V.C.); 4Department of Medicine and Surgery, University of Parma, Via Gramsci, 14, I-43126 Parma, Italy; federico.marchesi@unipr.it; 5Department of Biomedical and Neuromotor Sciences, University of Bologna, I-40126 Bologna, Italy; davide.gori4@unibo.it

**Keywords:** pneumococcus, *Streptococcus pneumoniae*, invasive pneumococcal disease, occupational disease, prevalence

## Abstract

Workers occupationally exposed to welding dusts and fumes have been suspected to be at increased risk of invasive pneumococcal disease (IPD). Since the 2010s, the United Kingdom Department of Health and the German Ständige Impfkommission (STIKO) actively recommend welders undergo immunization with the 23-valent polysaccharide (PPV23) pneumococcal vaccine, but this recommendation has not been extensively shared by international health authorities. The present meta-analysis was therefore designed to collect available evidence on the occurrence of pneumococcal infection and IPD among welders and workers exposed to welding fumes, in order to ascertain the effective base of evidence for this recommendation. PubMed, Embase and MedRxiv databases were searched without a timeframe restriction for the occurrence of pneumococcal infections and IPD among welders and workers exposed to metal dusts, and articles meeting the inclusion criteria were included in a random-effect meta-analysis model. From 854 entries, 14 articles (1.6%) underwent quantitative analysis, including eight retrospective studies (publication range: 1980–2010), and six reports of professional clusters in shipbuilding (range: 2017–2020). Welders had an increased likelihood of developing IPD compared with non-welders (odds ratio 2.59, 95% CI 2.00–3.35, I^2^ = 0%, *p* = 0.58), and an increased likelihood of dying from IPD (standardized mortality ratio (SMR) 2.42, 95% CI 1.96-2.99, I^2^ = 0%, *p* = 0.58). Serotype typing was available for 72 cases, 60.3% of which were represented by serotype 4, followed by 12F (19.2%) and serotype 8 (8.2%). Although the available data derive from a limited number of studies, available results suggest that pneumococcal vaccination should be recommended for workers exposed to welding fumes, and vaccination strategies should consider the delivery of recombinant formulates in order to combine the direct protection against serotypes of occupational interest with the mucosal immunization, reducing the circulation of the pathogen in occupational settings characterized by close interpersonal contact.

## 1. Introduction

*Streptococcus pneumoniae* or pneumococcus (PNX) is a highly invasive, Gram-positive, extracellular bacterial pathogen, and the diseases related to this pathogen represent a leading contributor to the global burden of vaccine-preventable morbidity and mortality [1,2,3,4]. PNX diseases range from non-invasive infections of the mucosa of the respiratory tract (i.e., otitis media and sinusitis) to severe conditions, in which the bacterium can be isolated from normally sterile sites (i.e., pneumonia, bacteremia, sepsis, meningitis and osteomyelitis), that are collectively reported as Invasive Pneumococcal Disease (IPD) [1,2,4,5].

Before the inception of the SARS-CoV-2 pandemic, *S pneumoniae* was historically acknowledged among the leading causes of mortality [2,5]. According to 2016 estimates, PNX pneumonia was the leading cause of lower respiratory infection morbidity and mortality with well over 1 million annual deaths, exceeding all of the other pathogens combined (e.g., 76,000 deaths for Respiratory Syncytial Virus, 58,000 deaths for seasonal influenza, and 48,000 deaths for *Haemophilus influenzae* type b) [6]. Usually, PNX infections affect people of all ages, with higher risk reported among infants and the elderly [5,6,7]. According to official data from the European Centre for Disease Prevention and Control (ECDC), between 2015 and 2019, a total of 114,787 IPD cases were reported across European Union (EU)/European Economic Area (EEA) countries, with the average case fatality ratio ranging between 14.5% and 17.4%. Even though the highest incidence rates occurred in infants in their first year of life (13.53 to 14.31 cases per 100,000 persons), and adults aged 65 years or older (16.15 to 18.73 cases per 100,000 persons), cases of IPD in working-age groups (25 to 64 years) represented a substantial share of the total disease burden, accounting to around 37% of all reported cases, with a case fatality ratio ranging between 19.2% and 23.7% in the age group of 45 to 64 years.

The usual risk factors for IPD and their severe outcomes in adults are identified as: abuse of alcohol; smoking history; chronic heart, lung, liver or renal disease; decreased immune function; functional or anatomic asplenia; diabetes; cochlear implants; and liquor fistulas [5,7]. Interestingly, an increased occurrence of PNX pneumonia and IPD among workers belonging to certain occupational groups has been repeatedly reported [8,9,10], particularly among welders and professionals exposed to welding fumes [11,12,13,14,15]. More precisely, a previous systematic review on work-related pneumococcal diseases identified an occupational burden equal to 10.0%, with an attributable fraction ranging between 38% and 70% [16]. Several likely explanations have been provided [10], mostly revolving around the inhalation of metals, particularly manganese and iron, that in turn would impair the pulmonary clearance of the pathogens. Not coincidentally, since October 2012 the British Department of Health has issued guidelines recommending that employers must plan PNX vaccination campaigns for employees exposed to welding or metal fumes. More precisely, pneumococcal polysaccharide vaccine (PPV) should be provided as a single 0.5 mL dose for occupationally exposed individuals, also taking into account the exposure control measures in place [17]. Similar recommendations have been shared since 2016 by the German Ständige Impfkommission (STIKO) from the Robert Koch Institut [18], being eventually included in the Austrian National Vaccination Plan [19]. As welders and workers exposed to welding and metal fumes are not consistently acknowledged among high-risk groups by other international health authorities, the systematic collection and review of the available evidence on pneumococcal disease in high-risk workers (i.e., those exposed to metal and welding fumes) could provide some significant insights as to the definition of more effective occupational vaccination strategies.

## 2. Materials and Methods

### 2.1. Study Selection, Inclusion and Exclusion Criteria

We performed the following systematic review and meta-analysis of the literature according to the “Preferred Reporting Items for Systematic Reviews and Meta-Analysis” (PRISMA) guidelines [20].

As a preliminary step, research concepts were defined according to the “PECO” strategy (i.e., Patient/Population/Problem; Exposure; Control/Comparator; Outcome) [21,22], as shown in Table 1. More precisely, the systematic review was designed to assess whether industrial workers (P), occupationally exposed to metal and welding fumes (E), compared to the general population or not occupationally exposed individuals (C), are affected or not by an increased occurrence of PNX pneumonia, IPD, and PNX-related deaths (O).

Relevant studies were searched for using 2 scholarly databases (PubMed/MEDLINE and EMBASE) and the pre-print repository medrxiv.org up to 28 February 2023, without backward chronological restrictions.

This search strategy utilized the combination of the following search strings, respectively, for PubMed (through Medical Subject Heading [MeSH] terms) and EMBASE:(a)(“*Streptococcus pneumoniae*” OR “pneumococcus” OR “pneumococcal infection”) AND (“meningitis” OR “pneumonia” OR “bacteremia” OR “invasive pneumococcal disease” OR “IPD”) AND (“occupation*” OR “work-related” OR “worker*” OR “job”)(b)(“*Streptococcus pneumoniae*”/exp OR “*Streptococcus pneumoniae*” OR “pneumococcal infection”) AND (“invasive pneumococcal infection” OR “meningitis” OR “infectious meningitis” OR “pneumonia” OR “IPD” OR “bacteremia”) AND (“occupation” OR “work” OR “workforce”)

Retrieved records were handled through the references management software Mendeley Reference Manager 2.84.0 (2023 Mendeley Ltd., London, UK) after having the title and abstract screened by two independent authors (FM and PF). Only original observational studies reporting the diagnosis of pneumococcal infection were included in the analyses: review articles, meta-analyses, meeting reports and conference abstracts were excluded from both qualitative and quantitative analysis. Articles that were consistent with the aims of the study were then text-reviewed to assess whether they met the following inclusion criteria:Reporting the crude number of assessed cases of PNX-related infections, IPD and/or PNX-related deaths: generic diagnoses such as “respiratory infections” or “pneumonia” not otherwise specified were removed from the analyses;Reporting the total number of exposed workers;Reporting the settings of occupational exposure, specifically focusing on the exposure to welding and metal fumes, assessed through job titles (e.g., welders) or job exposure matrix.Case reports, case series and outbreak reports were included but corresponding estimates were independently calculated.

Regarding exclusion criteria, we deliberately excluded: articles written in languages other than Italian, German, Swedish, English, French, Spanish and Portuguese (i.e., the languages spoken by the investigators).

### 2.2. Data Extraction

Data extracted included:Settings of the study (country, time, occupational settings);Number of included cases/deaths;Number of assessed workers;Number and characteristics of the reference groups (where available);Number of expected deaths for pneumococcal pneumonia in the reference group (where available);Pneumococcus serogroups, where available;Comorbidities, where available;Time elapsed between the first and the last case (where available).

### 2.3. Risk of Bias Assessment

After data extraction, the potential risk of bias of retrieved studies was assessed by means of the National Toxicology Program (NTP)’s Office of Health Assessment and Translation (OHAT) handbook and respective risk of bias (ROB) tool [23,24]. The property and design of the ROB tool have been otherwise described. In brief, the ROB tool assesses the internal validity of a given study, focusing on the study’s design and conduct, evaluating whether any bias has compromised the consistency of the link between the hypothesis (in this case, the exposure) and eventual outcome (in this case, IPD and/or PNX infections). The sources of potential bias that are investigated are: participant selection, confounding, attrition/exclusion, detection, selective reporting and other sources. During the assessment of the individual studies, researchers are then requested to rate the above items from “definitely low”, “probably low”, “probably high”, to “definitely high”. According to the OHAT handbook, even studies with “probably high” or “definitely high” ratings are included in the overall body of evidence, reviewed and eventually included in the meta-analysis. For the aims of the present review, the full texts of retrieved articles were rated according to the aforementioned indications by two independent investigators. Disagreements were then solved by consensus between the two reviewers; when it was not possible to reach consensus, input from a third investigator (M.R.) was searched and obtained.

### 2.4. Statistical Analysis

Descriptive analysis was initially performed by calculating the crude incidence rate, case-fatality ratio, and attack rates per 100 workers. Pooled estimates for PNX serogroups and comorbidities were also calculated.

In observational studies, pooled estimates for the standardized mortality rate (SMR) for PNX-related deaths, and the odds ratio (OR) for pneumonia and/or IPD in targeted groups compared to controls and/or the general population, were meta-analyzed through a random effect model (REM). The REM was preferred over the fixed-effect model to cope with the presumed heterogeneity in study design [25,26]. The inconsistency between included studies (i.e., the percentage of total variation across studies that could be associated with underlying heterogeneity rather than chance) was assessed through the calculation of the I^2^ statistic. I^2^ was interpreted as follows: 0% = no heterogeneity; 0% to 25% = minimal heterogeneity; 26% to 50% = mild heterogeneity; 51% to 75% = moderate heterogeneity; and >75% = strong heterogeneity [25].

Publication bias was then investigated through the calculation of contour-enhanced funnel plots, and the use of Egger’s test for quantitative publication bias analysis (at a 5% significance level). Radial plots were then calculated and visually inspected to rule out small study bias.

All analyses were performed by means of the “meta”, “metafor”, and “robvis” packages with R (version 4.0.3) [27] and RStudio (version 1.1.463) software. The aforementioned packages are open-source add-ons for conducting meta-analyses.

This systematic review was registered with the International Prospective Register of Systematic Reviews, or PROSPERO, with the progressive registration number CRD42023404926.

## 3. Results

### 3.1. Summary of Retrieved Studies

The flow chart of retrieved studies is provided in Figure 1. In brief, a total of 854 entries (including 99 from PubMed, 477 from MedRxiv, and 278 from EMBASE) were initially identified. A total of 343 articles were duplicated across the above databases and were therefore removed from the final analyses (40.2% of the initial pool). The remaining 511 entries were then screened by title and abstract, with the subsequent removal of 477 items (55.9%). The remaining 34 entries (4.0% of the initial sample) were full-text reviewed [8,9,10,11,12,13,14,15,16,28,29,30,31,32,33,34,35,36,37,38,39,40,41,42,43,44,45,46,47,48,49,50,51,52]. Of them, 20 were excluded from the final pool (2.3%), which only contained 14 articles (1.6% of the initial sample), whose content was included in both qualitative and quantitative analyses [8,13,29,30,32,33,34,35,37,38,41,42,43,47].

### 3.2. Summary of Case Series

In summary (see Table 2), six articles were reports on outbreaks of PNX-related diseases in occupational settings that included exposure to welding fumes [30,32,37,38,41,42]. Most reported outbreaks (four out of six) occurred in European shipyards between 2017 and 2020 [30,32,37,42]. The number of retrieved cases ranged from 3 in the small case series of Kiang [41] to 37 in the report from Cassir et al. [32], for a total of 110 IPD cases. All patients were male, with a median age range of 24–52 years. The total number of exposed workers was provided in five out of six case series [30,32,37,38,39], ranging from 50 [38] to over 7000 [42]. The interval between the first and last reported cases ranged from 29 days [32] to 209 days [42], with a median of 60.5 days. Corresponding attack rates ranged from 0.08 cases per 100 workers by month in the report by Linkevicius et al. [42], to 0.16 in the report by Ewing et al. [37], 0.51 in the report from Berild et al. [30], 0.66 in the study from Cassir et al. [32], and finally 8.00 in the study from Flodin [38]. Regarding the severity of reported cases, hospitalization rate ranged from 48.6% [32] to 100% [38,41], but even among larger samples, high hospitalization rates were reported: 30 out of 37 (81.1%) in the report from Linkevicius et al. [42], 18 out of 37 (48.6%) in the report by Cassir et al. [32], and 15 out of 20 in the study from Berild et al. [30]. Regarding admission to the ICU, reported rates ranged from 10.8% [32] to 25.0% [38], while, notably, only one death was reported, in the report by Linkevicius et al. [42], totaling a case fatality ratio of 2.7%. Identified serotypes were provided by four studies out of six, for a total of 73 cases out of 110 (66.4%), and the results are summarized in Table 3.

In brief, the most reported serotype was 4 (60.3%), followed by 12F (19.2%), 8 (8.2%), 3 and 9N (both 2.7%), while a single case was reported for serotypes 1, 9V, 14, 22F and 33F (1.4%). Interestingly, as represented in Table 3, all serotypes were included in the PPSV23, while PCV formulates were characterized by a relatively more limited potential protection, as all commercially available PCVs would fail in cases with serotype 9N (2.7%), PCV13 and PCV15 would fail in cases with serotypes 12F (19.2%) and 8 (8.2%), and PCV13 would also fail in cases with serotypes 22F and 33F (1.4% each).

### 3.3. Summary of Observational Studies

The remaining eight studies were observational study designs and were based on health records from national and local health authorities [8,9,29,33,34,35,47,48], as well as on the retrospective analysis of hospital records [34] (Table 4). Of the retrieved studies, three were based in the United Kingdom [33,34,35], three in Sweden [8,9,47], one in the USA [29], and one in Canada [13], and were performed on exposures and incident cases ranging from 1950 to 2019 (Figure 2). Retrieved studies were also heterogenous in terms of exposures and population, as six studies reported on the occurrence of PNX-related disorders in welders, identified through their job titles [8,9,13,29,33,35], while a job-exposure matrix was utilized in two further studies [34,47], that instead focused on the exposure to welding fumes. As a consequence, the assessed outcomes were also quite heterogenous: while the studies from Beaumont and Weiss [29], Coggon et al. [33] and Palmer et al. [35] provided an estimate for the standardized mortality ratio, the studies by Palmer et al. [34], Torén et al. [8,9] and Wong et al. [13] estimated the odds ratio for developing pneumococcal disease and/or IPD compared to a reference population, who were either unexposed or experienced more limited exposure to metal and welding fumes. Finally, the study by Torén et al. [47] provided an estimate of the risk ratio (RR) for developing IPD, which was calculated by means of Poisson regression analysis.

Following the characteristics of the study design, the collected samples were also quite heterogenous, with a sample size ranging from 8 deaths associated with IPD in the study by Torén in 2011 [47] to over 836 cases of IPD in the study by Torén et al. in 2022 [10].

Focusing on studies based on death registries [29,33,35], the sample of pneumococcal related deaths among welders ranged from 19 out of 410 total deaths (4.6%) [29] to 55 out of 97 (56.7%) in the report from Coggon et al. [33]. When dealing with reports on incident IPD cases, the number of index cases ranged from 18 to 49 [8,13,34]. However, characterization of occupational exposures was quite heterogenous, as the study by Palmer et al. did include welders and other metalworking professionals [34], while the study from Torén et al. encompassed welders and flame cutters [8], and conversely Wong et al. only included professional welders [13].

### 3.4. Risk of Bias Assessment

A detailed description of the ROB assessment has been summarized in Appendix A, Table A1 and Appendix A, Figure A1 for case series, and in Appendix A, Table A2 and Appendix A, Figure A2 for observational studies.

On the one hand, the majority of included case series studies were of high quality, being limitedly affected by potential bias. A notable exception was the report from Kiang [41], whose design lacked valuable information such as the total number of potentially exposed workers, which led to some possibility of residual selection bias. Moreover, some degree of outcome assessment and reporting bias also affected the studies from Berild et al. [30] and Linkevicius et al. [42], as both reports lacked key information about the eventual outcome of assessed cases; that is, the number of admissions to the ICU. Moreover, four out of the six studies [32,37,38,42] provided an analysis of potential risk factors and co-exposures, reducing the risk for confounding factors.

On the other hand, the overall quality of the observational studies was heterogenous. More precisely, the risk for selection bias (D1) was probably low in most of the cases, with only the reports from Palmer et al. [34] and Torén et al. [8] affected by some degree of selection bias, because of their respective inclusion criteria and the information source of included studies. On the contrary, more than half of the studies were reasonably affected by some degree of bias in terms of exposure assessment [13,29,33,35], as it was only defined in terms of exposure, while the remaining studies performed a more accurate estimate by means of detailed job titles. In terms of outcome assessment, the collected studies were mostly either scarcely or unlikely affected by any bias, even though the older study from Beaumont and Weiss [29] and the report from Torén et al. [47] were presumably affected by substantial or likely high bias, because of their lack of specific diagnoses of pneumococcal infections and IPD in the parent information sources. Unfortunately, all articles were lacking accurate analyses of potential confounding factors represented by comorbidities (D4) and potential co-exposures (D6), with substantially low estimates for both variables. Finally, half of the included studies were reasonably affected by some degree of reporting bias (D5): while studies based on death registries were unlikely to fail in retrieving cases associated with the inquiry topic (i.e., IPD and pneumococcal infections), the studies based on notification reports were necessarily affected by some degree of inaccuracy.

### 3.5. Meta-Analysis of Case Series Studies

As shown in Figure 3, meta-analysis was performed on five out of six case series, as the report from Kiang et al., lacked an estimate of the total number of workers potentially exposed to PNX. The notification rate for PNX was estimated to be 0.853 per 100 workers (95% CI 0.359 to 2.013) (Figure 3a), while the hospitalization rate was estimated to be 0.068 cases per 100 workers (95% CI 0.169 to 2.599) (Figure 3b). The pooled admission rate was calculated over three studies and estimated to be 0.068 (95% CI 0.030 to 0.150), as the reports from Berild et al. [38] and Linkevicius et al. [42] did not provide this information. Finally, with only one death reported from all the studies, an eventual death rate of 0.006 per 100 workers (95% CI 0.001 to 0.040) was calculated.

When dealing with attack rate (Figure 4), an estimate of 4.22 cases per 100 persons- years (95% CI 0.86 to 7.58) was eventually calculated.

With the exception of the death rate (I^2^ = 0%, *p* = 1.000), estimates for heterogeneity (I^2^) were consistently substantial, ranging from 81% for the ICU admission rate, 89% for the notification rate, 92% for the attack rate, to 95% for the hospitalization rate.

### 3.6. Meta-Analysis of Observational Studies

Estimates of SMR were calculated over three studies with a total of 712 deaths [29,33,35], 106 of them (14.9%) occurring because of pneumococcal infections in professional welders. Conversely, the large prospective study from Torén et al. [47] was excluded from pooled analyses as the authors considered exposure to metal fumes and welding tasks within a broader range of exposures associated with the construction industry. Interestingly, more than half of the included cases were retrieved from a single study [33]. A pooled estimate of 2.42 (95% CI 1.96 to 2.99) was eventually calculated. Interestingly, the heterogeneity among the assessed studies was very low (I^2^ = 0%, *p* = 0.58) (Figure 5).

The association of IPD cases in welders compared to workers not exposed to metal fumes was estimated through the calculation of the corresponding ORs and their respective 95% CIs, as shown in Figure 6. In brief, a total of three studies, with a total of 89 cases, were included in the analyses [8,13,34]. Again, the large study from Torén et al. [9] was not included in the pooled estimates because of the exposure assessment, which was based on a job exposure matrix and not consistent with the other reports [8,13,34]. Nonetheless, while the study by Wong et al. [13] only reported corresponding estimates regarding professional welders, both Palmer et al. [34] and Torén et al. [8] included in their reports estimates for other metalworkers and/or flame cutters. Moreover, more than half of the included cases were retrieved from a single study; that is, the report by Torén et al. [8]. A pooled OR of 2.59, with a 95% CI 2.00 to 3.35, was eventually calculated, with very low heterogeneity (I^2^ = 0%, *p* = 0.58).

### 3.7. Analysis of Publication Bias

Potential publication bias was evaluated by means of the calculation of funnel plots, while potential small study bias was assessed by means of radial plots. The funnel plots are reported in Appendix A, Figure A3, while the corresponding radial plots are presented as Appendix A, Figure A4, and the detailed results of the Egger’s test are summarized in Appendix A, Table A3. In funnel plots, the effect size of every study is plotted against the corresponding estimate of standard error. The asymmetrical distribution of each point at visual inspection is considered suggestive of publication bias (i.e., publication of the study depending not just on the quality of the research, but also on the hypothesis tested, and the significance and direction of detected effects). Despite a cautious appraisal required by the reduced number of included studies, the large majority of estimates were affected by some degree of publication bias. However, such subjective evidence from the funnel plots was not confirmed by the regression test. In fact, the Egger’s test ruled out publication bias for most of the included findings, with the notable exception of the attack rate (Appendix A, Figure A3e; intercept = 3.586, SE = 1.004, t = 3.57; *p* = 0.038). On the other hand, in the radial plots (Appendix A, Figure A4), corresponding estimates were somehow scattered across the regression line: despite the reduced number of samples collected, a small study effect for these findings could be therefore ruled out.

## 4. Discussion

In this systematic review and meta-analysis about IPD and pneumococcal diseases among workers exposed to welding and metal fumes, we were able to retrieve a total of fourteen studies, six case series, and eight observation studies.

The information conveyed by the collected studies allowed some estimates of the actual occurrence of IPD among welders, even though most of them were collected from a very specific sub-setting, i.e., shipyards. In a pooled population of 17,673 workers and 107 IPD cases, an attack rate of 0.34 per 100 workers per month was eventually estimated, with a notification rate of 0.853 per 100 workers, a hospitalization rate of 0.667 per 100 workers, and a death rate of 0.006 deaths per 100 workers, as, in fact, only one death was eventually reported. The estimates of the severity of the IPD, ascertained through the proxy of ICU admissions, were calculated using a reduced population (i.e., 8873 workers), with a pooled estimate of 0.068 cases per 100 workers.

When taking into account the aforementioned figures, it should be stressed that, despite vaccination programs [1,6,28,53], the burden of disease associated with PNX in the general population still remains significant. For example, Palmborg et al., in their longitudinal study based on 10 years of public surveillance data in Nordic Countries (i.e., Denmark, Finland, Norway and Sweden), provided annual incidence rates ranging between 31.4 and 41.8 per 100,000 [54]. Similarly, Ochoa-Gondar et al. recently estimated a global incidence rate of 90.7 cases of pneumococcal pneumonia per 100,000 persons-year [55]. The outbreak potential of PNX, both at community level and in specific settings (i.e., schools, military and hospitals) has been previously highlighted [56], and occupational case studies collectively suggest that, in the specific settings of construction and naval shipyards, significant attack rates and eventual incidence estimates could be reached, representing a substantial public health issue.

Several explanations could be provided, as previously summarized by Toren et al. [8,9,10,47,48], relying not only on the characteristics of assessed workplaces and related exposures, but also on the characteristics of PNX and more general risk factors [2,5,56]. On the one hand, welders are occupationally exposed to several factors, including welding fumes, that elicit chronic damage to upper and lower airways, increasing the risk of invasive infections from respiratory pathogens, including PNX [45]. On the other hand, there is considerable evidence that the health status of professionals involved in naval and construction yards is often affected by a high occurrence of behavioral risk factors, including smoking habits and high alcohol consumption, that are in turn often associated with obesity, diabetes mellitus and chronic respiratory conditions [1,5,48,54,56]. Moreover, housing conditions should be considered. In fact, personnel from construction yards and shipyards often include a high share of workers who live in precarious and overcrowded settings, shared with workers from the same employer [57,58,59]. In these settings, the transmission of pathogens, such as meningococcus and SARS-CoV-2, has been extensively documented [60,61,62,63], and a summary for PNX outbreaks occurring between 1977 and 2017 confirms that the multiple modes of transmission, not necessarily limited to the nasopharyngeal carriers, were collectively related to close interpersonal contact [2,56]. Not coincidentally, some reports have documented an increased occurrence of IPD in settings such as mines and extractive industry, where the exposures to welding fumes are limitedly documented, but the workforce shares very similar specificities in terms of health status and housing issues [49,51], and also in close contacts during their daily tasks, with high sharing of devices and surfaces. The increased risk for professionals exposed to welding fumes is otherwise confirmed by retrospective studies, where an OR of 2.59, with a 95% CI of 2.00 to 3.35, was eventually calculated [8,13,34,44].

Interestingly, while the case series suggested that IPD associated with the occupational exposure to welding fumes would cause a very limited mortality, as shown by the very low case fatality ratio, the observational studies provided a substantially increased estimate for SMR (2.42, 95% CI 1.96 to 2.99) [29,33,35]. In other words, through a retrospective approach, a substantially higher mortality risk for welders and those exposed to welding fumes, following IPD infections, was the result. However, when dealing with these estimates and comparing them to those from case series, several main caveats should be preventively taken into account.

First and foremost, among the eight observational studies we were able to retrieve, a heterogenous approach was identified in terms of case definition and assessed outcomes. For example, while the study from Beaumont and Weiss [29] retrospectively reported all cases of mortality, and we identified IPD cases by means of the death certificates that in turn were coded according to the seventh revision of the International Classification of Diseases, other reports were based on more accurate reporting systems [8,9,13,34,47], that specifically included IPD as a reported disease or cause of death. Moreover, two large and high-quality studies from Torén et al. [9,47], were based on a heterogeneous strategy for the characterization of occupational exposures. As a consequence, only six reports were eventually included in the quantitative analyses, and the calculation of a pooled SMR for pneumococcal infections and a pooled OR for IPD was derived from three studies each.

Second, because of their retrospective design strictly based on health registries, several included studies were unable to provide a series of key information, such as the serogroups associated with assessed PNX infections, baseline comorbidities and individual risk factor—most notably, smoking history. The most notable information gap is clearly represented by the vaccination status of the affected individuals. PPSV was made available in the 1970s as a 14-valent formulate, that then evolved into the current 23-valent formulations available since 1983 [1,53], and it has been shown as quite effective in reducing the risk of developing PNX-related illnesses, including IPD, if delivered before the first exposure [56]. For example, Zivich et al. estimated a vaccine effectiveness of 87% (95% CI −3% to 98%) for outbreaks occurring before the availability of conjugated formulations. As we cannot rule out that a significant share of the individuals included in the estimates collected after the late 1980s did include a substantial share of subjects previously vaccinated with PPSV23, the corresponding attack rate and IPD related morbidity and mortality could underestimate the potential severity of PNX-related illness occurring in unvaccinated welders.

Third, as otherwise summarized in Figure 2, all of the retrospective studies included occupational exposures that began in a very distant timeframe. The study by Beaumont and Weiss included occupational exposures that started in 1950 [29], while the studies from the UK and Nordic countries, even though they reported on mortality from a more recent period, included professionals with professional exposure that reasonably began in the late 1970s [8,9,10,47,48]. As a consequence, all of these estimates should be considered as the late consequences of health and safety requirements for workplaces that are hardly comparable to the case series, whose landscapes are restricted to the last decade. In this regard, the report accuracy may have been affected by the heterogenous approach to disease reporting across years and different countries, introducing even greater heterogeneity in the resulting estimates. Moreover, the more severe consequences of IPD when diagnosed in the community are usually associated with older age groups [53,54,55], and the majority of the retrospective studies by their design include large populations of retired and older professionals. This is hardly comparable to active workers, even when focusing on “older workers”; that is, occupationally active individuals aged 55 years or more [64,65,66]. Therefore, both the increased risk and mortality associated with IPD should be more correctly interpreted through the lens of a targeted population who are particularly vulnerable.

Even though only limited recommendations for PNX vaccination in welders have been issued, our results collectively stress the preventive value of this intervention, from a public health point of view. Again, some considerations should be taken into account. First of all, to date, the large majority of WHO member states (i.e., 148 out of 194) have introduced PCV into their National Immunization Programs for infants and children, either nationally or sub-nationally, and these interventions have largely involved both infants and adults [5,67,68,69,70]. On the other hand, the increasingly older workforce of Western countries will benefit from vaccination strategies for older adults, including both PPSV23 and PCV. In a relatively brief time, the adult workforce would therefore encompass a large number of subjects with some degree of immunization against PNX, being therefore protected against its more severe complications. Second, IPDs are associated with a large number of different serogroups, and therefore an accurate assessment of vaccination strategies forcibly requires the preventive identification of the strains associated with occupational settings [2,56]. More precisely, vaccination strategies for IPD will be highly effective if the serogroups associated with occupational infections (and particularly with occupational outbreaks) are caused by strains included in the vaccines [68,70]. From this point of view, not only are the available studies highly defective but also somehow inconsistent with other reports [56]. In fact, serogroup data were only available from some of the case series, specifically a total of 73 cases (66.4%), and the majority of them were associated with serogroup 4 (60.3%), with a notable representation of serogroup 12F (19.2%) and 8 (8.2%); conversely, the study by Zivich et al. [56] did stress a main role for serogroups 9 (15.4%), 1 (15.4%), 23F (12.8%), and serotype 14 (12.8%). As a consequence, the available data suggest that only some vaccine formulates would guarantee an effective protection for the whole of potential strains, most notably the PPSV23 and PCV20 vaccines. However, PPSV and PCV have quite distinctive profiles, not only in terms of recommendations, but also when dealing with the characteristics of the elicited immunity [68,69,70]. Nowadays, available occupational recommendations point towards the use of PPSV23 for older workers who have not completed a previous vaccination schedule with PCV. Nonetheless, the cheaper PPSV23 would only guarantee effective protection for the vaccinated individual, as it would not affect the circulation of the pathogen. However, this strategy has been proven effective, even in outbreak settings, as suggested by the report by Cassir et al. [32]. On the contrary, as stressed by Berild et al. [30], PCV vaccines, particularly PCV20, elicit mucosal immunity that in turn impairs the inter-human transmission of PNX, eventually reducing the risk of outbreaks, particularly in those settings (such as construction yards and naval shipyards) where a continuum between workplaces and households increased the risk for the person-to-person spreading of the pathogen in crowded workplaces and through shared surfaces [28,30,32,37]. While a vaccination strategy based on PPSV would clearly fulfill the primary aim of occupational health practices, in terms of safeguarding and promoting the health of workers [71], the extensive use of PCV could provide substantial assistance in the containment of the PNX burden of disease in the general population, but only through vaccination campaigns based on more comprehensive formulates (e.g., PCV20).

### Limits

Despite the potential interest for Public Health and Occupational Health professionals, our study is affected by some limitations.

Firstly, meta-analyses are highly dependent on the quality of the original studies [72,73], and potentially affected by their high heterogeneity [73]. In this case, the quality of the studies we were able to retrieve was highly heterogenous, as stressed by the ROB assessment; in particular, observational studies were affected by significant shortcomings such as the case definition, the definition of occupational groups and exposure and the very large timeframe of the sample collection. Moreover, the estimates were based on a reduced number of populations, particularly when we pooled the data from observational studies. In fact, even the analyses we performed in order to rule out publication and small study effect should be very cautiously assessed.

Secondly, most of the collected evidence from observational studies was based on only two countries; that is, Sweden and the United Kingdom. Even though both settings are characterized by highly regulated occupational health frameworks, and highly developed healthcare systems, the corresponding features are only limitedly comparable to most, not all, high-income countries. The different background epidemiology of the pathogen in the general population, the baseline health status of occupational groups and the workplace exposure standards for welders and metal fumes collectively introduce significant heterogeneity to all pooled estimates, and we recommend a precautionary approach when assessing our eventual results.

Thirdly, the present study is based on reports that have been published since the early 1980s [29], that in turn include cases reported since the 1950s, with resulting heterogeneity in diagnostic criteria. This is particularly significant when dealing with pneumococcal pneumonia. While the diagnosis of IPD is relatively reliable as it insists on the identification of pneumococcal infection in a sterile site [3,7], earlier studies on the mortality associated with pneumococcal pneumonia were based on death certificates [29,33,35], that in turn included causes of death according to the International Classification of Diseases (7th edition for the study by Beaumont et al., [29]; 9th edition for the studies by Palmer et al. [35], and Coggon et al. [33]). While the report from Beaumont et al. [29] is unclear as to whether any microbiological analysis of respiratory specimens was actually performed, the studies from Coggon et al. [33] and Palmer et al. [35] specifically included the diagnosis of pneumococcal pneumonia, but a noticeable number of cases (at least 8 out of 55 cases in the series from Coggon et al. [33]) did not receive a coroner examination. As a consequence, we cannot rule out a significant overestimation of the actual mortality associated with pneumococcal pneumonia, particularly from older studies.

Finally, our estimates on the mortality associated with pneumococcal pneumonia were affected by the inconsistency of the high-quality study from Torén et al. [9,47] compared with other reports when dealing with the exposures and occupational settings. While other studies reported their estimates on professional welders [29,33,35], the report from Torén et al. [47] included a total of eight deaths associated with pneumococcal pneumonia (RR 5.77, 95% CI 1.53 to 21.73) which occurred in construction industries. In other words, the sample included a pool of workers exposed to a broad range of occupational respiratory risk factors, with extensive overlap with inorganic dusts, chemicals, and wood dust. Similarly, the study on IPD from Torén et al. [9], based on Swedish registries, reported an increased occurrence of this condition among workers exposed to metal fumes (OR 2.24, 95% CI 1.41 to 3.35, calculated by means of logistic regression analysis), but again it should be stressed that such estimates were not specifically calculated on professional welders, but rather on workers exposed to welding fumes in a broader range of occupational tasks.

## 5. Conclusions

In conclusion, the IPD burden among welders and workers exposed to welding fumes is substantial. Even though welding fumes include factors that can damage upper and lower airways, increasing the risk for both PNX infections and their complications, the causes are not reasonably limited to the occupational exposures. Unfortunately, the design of the available studies often impairs an accurate appraisal of occupational and non-occupational risk factors in the definition of increased occurrence and mortality for IPD in this occupational group. However, our results collectively stress the potential significance of PNX vaccination strategies in occupational settings. In this regard, actual evidence suggests that PPSV23 could represent a likely and effective option, but updated PCV (including PCV20) vaccines should be considered for a more comprehensive preventive strategy.

## Figures and Tables

**Figure 1 vaccines-11-01495-f001:**
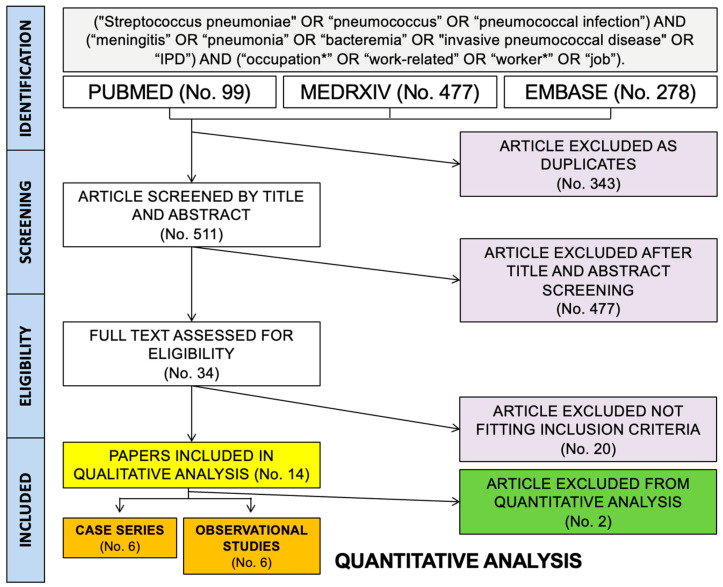
Flowchart of included studies.

**Figure 2 vaccines-11-01495-f002:**
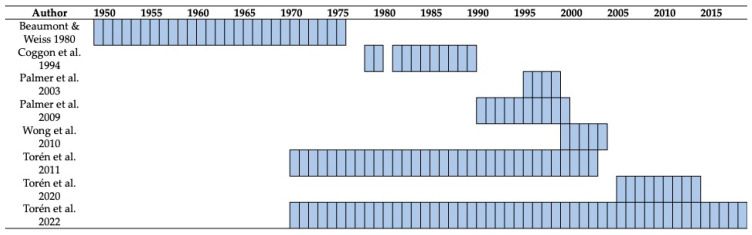
Timeframe of the collected studies [8,9,13,29,33,34,35,47].

**Figure 3 vaccines-11-01495-f003:**
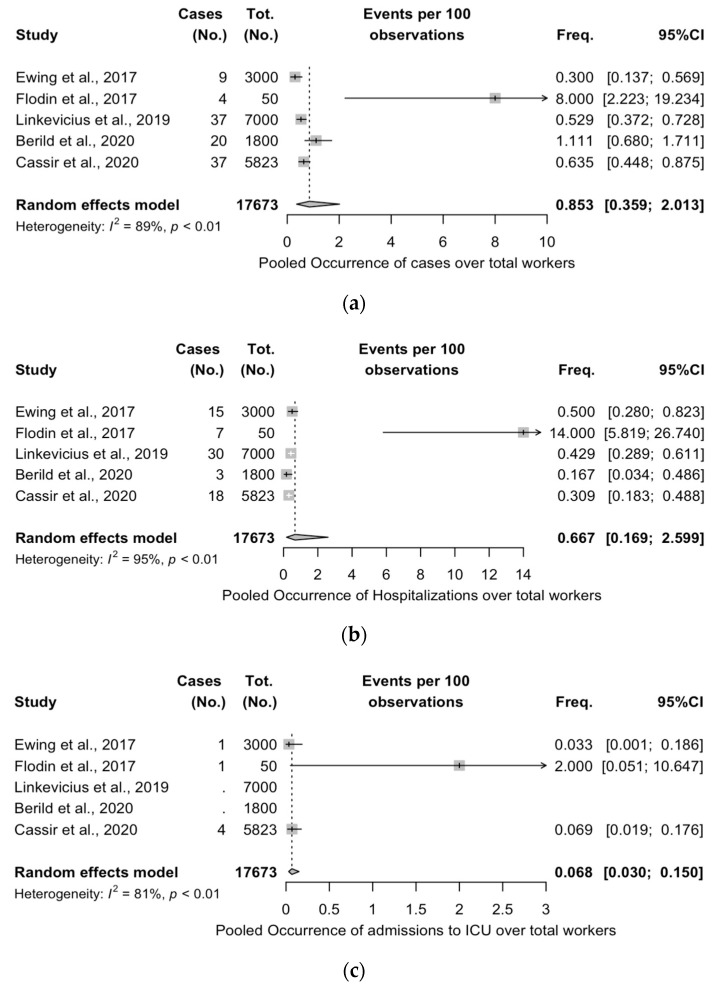
Forest plot for notification rate (**a**), hospitalization rates (**b**), admission to Intensive Care Unit (ICU) (**c**), and deaths (**d**) as collected from 5 out of 6 case series included in the present study [30,32,37,38,42].

**Figure 4 vaccines-11-01495-f004:**
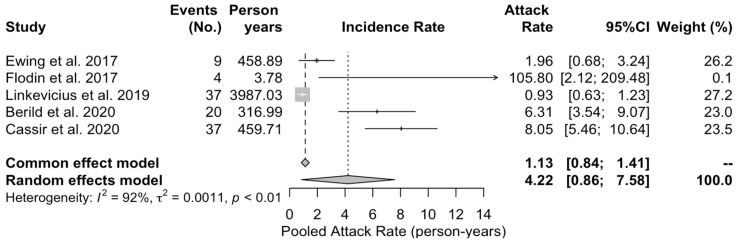
Forest plot of pooled attack rate for pneumococcal pneumonia (PNX) and invasive pneumococcal diseases in welders. Overall, an attack rate equal to 4.22 per 100 person-years, 95% CI 0.86 to 7.58, was calculated through a random effect model, with a very high heterogeneity (I^2^ = 92%) [30,32,37,38,42].

**Figure 5 vaccines-11-01495-f005:**
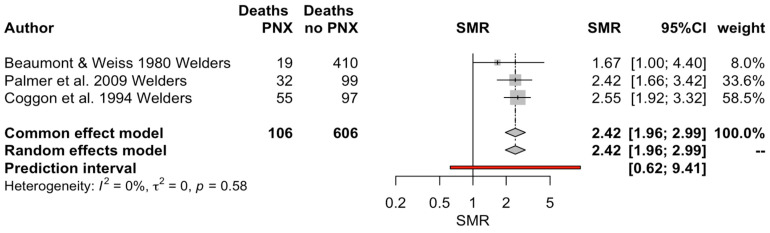
Forest plot of pooled standardized mortality ratio (SMR) for pneumococcal pneumonia (PNX) and invasive pneumococcal diseases in welders. Overall, a pooled SMR equal to 2.42, 95% CI 1.96 to 2.99, was calculated, with a very low heterogeneity (I^2^ = 0%) [29,33,35].

**Figure 6 vaccines-11-01495-f006:**
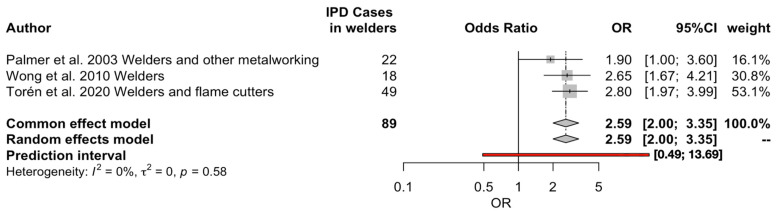
Forest plot of pooled odds ratio (ORs) for invasive pneumococcal diseases (IPD) in welders compared to cases occurring in non-welders. Overall, a pooled OR equal to 2.59, 95% CI 2.00 to 3.35, was calculated, with a very low heterogeneity (I^2^ = 0%) [8,13,34].

**Table 1 vaccines-11-01495-t001:** PECO worksheet [21,22].

Item	Definition
Population of interest	Among industrial workers, what is the effect of
Exposure	Occupational exposure to metal and welding fumes versus
Control/Comparator	General population and/or workers not occupationally exposed to metal and welding fumes in the
Outcome	Occurrence of PNX pneumonia, IPD, and PNX-related deaths

**Table 2 vaccines-11-01495-t002:** Characteristics of cohort studies included in the systematic review (Note: ICU = intensive care unit).

Authors	Year	Year of Cluster	Country	No. Cases (No., %)			No. Exposed Workers (No.)	Age (Years)		Hospitalizations (No., %)	ICU (No., %)	Deaths (No., %)	Serotypes (No., %)	Interval (Days)	Attack Rate (Cases per 100 Workers/Month)
				Confirmed	Probable	Total *		Median	Range						
Cassir et al. [32]	2020	2020	France	18, 48.6%	19, 51.4%	37, 0.6%	5823	39	22 to 66	18, 48.6%	4, 10.8%	0, -	3 (1, 11.1%) 4 (5, 55.6%) 8 (2, 22.2%) 9N (1, 11.1%)	29	0.66
Linkevicius et al. [42]	2019	2019	Finland	31, 83.8%	6, 16.2%	37, 0.5%	7000	48	19 to 64	30, 81.1%	n.a.	1, 2.7%	12F (13, 52.0%) 4 (11, 44.0%) 8 (1, 4.0%)	209	0.08
Kiang [41]	2018	-	Singapore	3, 100%	0, -	3	n.a.	24	20 to 41	3, 100%	3, 100%	0, -	n.a.	90	n.a.
Berild et al. [30]	2020	2019	Norway	10, 50%	10, 50%	20, 1.1%	1800	47	20 to 60	15, 75.0%	n.a.	0, -	4 (17, 100%)	65	0.51
Ewing et al. [37]	2017	2015	Northern Ireland	4, 44.4%	5, 55.6%	9, 0.3%	3000	43	20 to 60	7, 77.8%	1, 11.1%	0, -	3 (1, 25.0%) 4 (3, 75.0%)	56	0.16
Flodin et al. [38]	2017	2015	Sweden	4, 100%	0, -	4, 8.0%	50	52	37 to 58	4, 100%	1, 25.0%	0, -	n.a.	30	8.00

* percent value calculated on the total of exposed workers.

**Table 3 vaccines-11-01495-t003:** Serotypes of retrieved cases compared to the coverage of main available vaccines (i.e., PPSV23 = Pneumococcal polysaccharide vaccine; PCV = Pneumococcal conjugate vaccine).

				Serogroup																				
	1	2	3	4	5	6A	6B	7F	8	9N	9V	10A	11A	12F	14	15B	17F	18C	19A	19F	20	22F	23F	33F
PPSV23	X	X	X	X	X	X	X	X	X	X	X	X	X	X	X	X	X	X	X	X	X	X	X	X
PCV20	X		X	X	X	X	X	X	X		X	X	X	X	X	X		X	X	X		X	X	X
PCV15	X		X	X	X	X	X	X			X				X			X	X	X		X	X	X
PCV13	X		X	X	X	X	X	X			X				X			X	X	X			X	
Cases (No, %)	1, 1.4%		2, 2.7%	44, 60.3%					6, 8.2%	2, 2.7%	1, 1.4%			14, 19.2%	1, 1.4%							1, 1.4%		1, 1.4%

**Table 4 vaccines-11-01495-t004:** Summary table of observational studies on pneumococcal diseases (PD) in occupational settings. Notes: IPD = invasive pneumococcal diseases; OBS = observed; SMR = standardized mortality ratio; OR = odds ratio; RR = risk ratio; 95% CI = 95% confidence interval; M.A. = included in the meta-analysis.

Authors	Year	Timeframe	Country	Design	Settings	Exposure Assessment	Obs. (Welders) (No.)	Results	M.A.
Beaumont and Weiss [29]	1980	1950 to 1973	USA (Washington)	Analysis of death certificates	Metal trade workers in the area of Greater Seattle (≥3 years of seniority); original sample size of 3247 welders	Job title (Welders)	19	SMR 1.67, *p* < 0.001 for welders compared for pneumococcal pneumonia	Y
Coggon et al. [33]	1994	1979 to 1980; 1982 to 1990	United Kingdom	Analysis of death certificates	Nationwide data from official registries of metal-working occupations (occupational mortality data of Registrar General for England and Wales); original sample size of 729 death files.	Job title (Welders)	55	SMR 2.55, 95% CI 1.92 to 3.32 for pneumococcal pneumonia	Y
Palmer et al. [35]	2009	1991 to 2000	United Kingdom	Retrospective, based on death certificates	Nationwide data retrieved from the UK Office of Statistics, men aged 16 to 74 years from England and Wales. Original sample size of 794 death files.	Job title (Welders)	32	SMR 2.92, 95% CI 0.79 to 2.41 for pneumococcal pneumonia	Y
Palmer et al. [34]	2003	1996 to 1999	United Kingdom	Incident IPD cases, based on hospital records	Residents aged 20 to 64 years (650,000 individuals) from five metropolitan districts of the West Midlands. Admission to any of the acute medical services of the area. Original sample size, 525 cases of pneumonia.	Occupational exposure to welding fumes by detailed job titles	22	OR 1.8, 95% CI 0.6 to 5.2 for exposure to welding fumes in cases compared to control	Y
Torén et al. [47]	2011	1971 to 2003	Sweden	Mortality from infectious bacterial pneumonia, including pneumococcal pneumonia	Swedish Construction Workers from Swedish Construction Industry’s Organization for Working Environment; causes of death retrieved from national Cause of Death Register. Original sample of 773 cases of deaths from IPD.	Occupational exposure to metal fumes derived from the analysis of detailed job titles	8	RR 5.77, 95% CI 1.53 to 21.73 (calculated by means of Poisson’s regression analysis) for exposure to metal fumes compared to reference group.	N
Torén et al. [9]	2022	2006 to 2019	Sweden	Incident IPD cases, based on official Swedish registry	Swedish national Hospital Discharge Registry, identification of IPD. Controls retrieved from Swedish National Population registry. Original sample size of 3184 cases with pneumonia.	Occupational exposure calculated through a job exposure matrix based on the job title.	836	OR 2.24, 95% CI 1.41 to 3.35, calculated by means of logistic regression analysis for IPD in any exposure from metal fumes.	N
Torén et al. [8]	2020	2006 to 2014	Sweden	Incident IPD cases, based on official Swedish registry	Swedish National Hospital Discharge Registry, identification of IPD. Controls retrieved from Swedish national population registry. Original sample size of 4438 cases of IPD.	Occupational exposure calculated through the detailed job title.	27	OR 2.99, 95% CI 2.09 to 4.30 for welders.	Y
Wong et al. [13]	2010	2000 to 2004	Canada (Alberta)	Incident IPD cases, based on official registries	Official notification data on laboratory-confirmed IPD from the Canadian state of Alberta. Original sample size of 1768 IPD cases from a working population of around 3,000,000 inhabitants.	Job title (Welders)	18	OR 2.653, 95% CI 1.670 to 4.215 for working as welder compared to the general population	Y

## Data Availability

Raw data are available on request to the corresponding author.

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
