# Peer review of "Vaccinating Welders against Pneumococcus: Evidence from a Systematic Review and Meta-Analysis"

_vaccines, 2023, doi:10.3390/vaccines11091495_

Round 1

Reviewer 1 Report

This was a systematic review and meta-analysis designed to evaluate the occurrence of pneumococcal infection and IPD in welders and workers exposed to welding fumes, which would help determine whether pneumococcal vaccination should be offered to these workers. The conclusions derived from the findings were that although the data were derived from a limited number of studies, the data suggests that pneumococcal vaccination should be offered to such workers. 

COMMENTS

1) There is a significant typographical error in Figure 1. In the column going down from "PUBMED" the second last rectangle should read "PAPERS INCLUDED IN QUALITATIVE ANALYSIS (No. 14)"

2) It did not appear to me that figure 2 was referred to in the text. 

3) Another potential limitation of this study is that while the diagnosis of "IPD" would be clear in most studies (pneumococcal infection in a sterile site) the diagnosis of "pneumococcal pneumonia" would be based on the interpretation of microbiological findings by the authors of the studies (e.g., based on sputum culture). In that respect do the authors know how the latter diagnosis was made? 

There are a few grammatical errors, which if corrected would improve the manuscript. 

Author Response

Estimated Reviewer,

thank you for your accurate and collaborative review.

We've amended the text according to your recommendations, and more precisely:

1) There is a significant typographical error in Figure 1. In the column going down from "PUBMED" the second last rectangle should read "PAPERS INCLUDED IN QUALITATIVE ANALYSIS (No. 14)"

ANSWER: Thank you for your note; we've amended the text and modified the figure accordingly.

2) It did not appear to me that figure 2 was referred to in the text. 

ANSWER: Thank you for your note. Figure 2 is now referred into the text at page 2 and page 16 (row 462)

3) Another potential limitation of this study is that while the diagnosis of "IPD" would be clear in most studies (pneumococcal infection in a sterile site) the diagnosis of "pneumococcal pneumonia" would be based on the interpretation of microbiological findings by the authors of the studies (e.g., based on sputum culture). In that respect do the authors know how the latter diagnosis was made? 

ANSWER:

Thank you for your note. Indeed, we agree that the potential limit you've addressed must be reported and discussed in the revised text. We've amended the text as follows:

"

Third, the present study is based on reports that have been published since the early 1980s [29], that in turn included cases reported since the 1950s, with resulting heterogeneity in diagnostic criteria. This is particularly significant when dealing with pneumococcal pneumonia. While the diagnosis of IPD is relatively reliable as it insists on the identification of pneumococcal infection in a sterile site [3,7], earlier studies on the mortality associated with pneumococcal pneumonia were based on death certificates [29,33,35], that in turn included causes of death according to the International Classification of Diseases (7th  edition for the study of Beaumont et al., [29]; 9th edition for the studies of Palmer et al. [35], and Coggon et al. [33]). While the report from Beaumont et al. [29] is unclear whether any microbiological analysis of respiratory specimens was actually performed, the studies from Coggon et al. [33] and Palmer et al. [35] specifically included the diagnosis of pneumococcal pneumonia, but a noticeable number of cases (at least 8 out of 55 cases in the series from Coggon et al. [33]) did not receive a coroner examination. As a consequence, we cannot rule out a significant overestimation of the actual mortality associated with pneumococcal pneumonia, particularly from older studies."

Eventually, we again thank you for your collaborative review, and we hope that the amended paper could be accepted for pubblication on Vaccines.

Reviewer 2 Report

Major comments: The present systematic review and meta-analysis investigated the increased occupational risk of welders in developing invasive pneumococcal disease (IPD) and dying from IPD by including 8 retrospective cohort studies and 6 case series of reports of professional clusters in shipbuilding. As there is no doubt that welders should receive pneumococcal vaccination, this systematic review provides good evidence for an occupational indication for pneumococcal vaccination. However, I recommend that the authors clarify two important points:

1.) In the abstract (line 19-20) and the introduction (line 80-81) the authors claim that only health authorities in the UK recommend welders immunization against pneumococcus - which is not true. At least the STIKO (Standing Committee on Vaccination at the Robert Koch Institute) in Germany recommends vaccinating of welders with PPV23 since 2016 and also the current Austrian vaccination schedule defines welders as high risk population for invasive pneumococcal disease.

2.) In the abstract and the main text the authors describe that they included 8 retrospective observational studies. As shown in Table 4 they did not include the prospective cohort study by Toren et al. Thorax 2011 - what was the reason for not including the only large prospective cohort study in the Meta-analysis?

Minor comments: 

Introduction Line 65: The authors should also include cochlear implants and liquor fistulas as risk factors for IPD

Table 3: The authors should define the abbreviations Conf. and Prob.

Author Response

Estimated Reviewer,

first of all, thank you for the accurate and collaborative review you've provided. We've amended the text according to your recommendations, and we're confident that the improvements we've performed have eventually increased the readability and the significance of this study.

More precisely:

1.) In the abstract (line 19-20) and the introduction (line 80-81) the authors claim that only health authorities in the UK recommend welders immunization against pneumococcus - which is not true. At least the STIKO (Standing Committee on Vaccination at the Robert Koch Institute) in Germany recommends vaccinating of welders with PPV23 since 2016 and also the current Austrian vaccination schedule defines welders as high risk population for invasive pneumococcal disease.

ANSWER:

Thank you! In fact, we did miss the improved STIKO recommendations, and those from Austria as well. The paper has been amended as follows:

a) abstract: Since 2010s, the United Kingdom Department of Health and the German Ständigen Impfkommission (STIKO) actively recommends welders immunization with the 23-valent polysaccharide (PPV23) pneumococcal vaccine, but this recommendation has not been extensively shared by international health authorities

b) main text: Similar recommendations have been shared since 2016 by German Ständigen Impfkommission (STIKO) from the Robert Koch-Institut [18], being eventually included in the Austrian National Vaccination Plan [19]. 

2.) In the abstract and the main text the authors describe that they included 8 retrospective observational studies. As shown in Table 4 they did not include the prospective cohort study by Toren et al. Thorax 2011 - what was the reason for not including the only large prospective cohort study in the Meta-analysis?

ANSWER:

Thank you for your note. In fact, we've removed from pooled analyses two studies from Torén et al even though their size and quality would have substantially contributed to our study. The reason is associated with the exposure assessment strategy from the aforementioned studies. In fact, while all other studies focused on job titles and actual tasks, Torén et al. ascertained their exposure assessment by means of a job exposure matrix. In other term, albeit reasonably more accurate, this approach encompassed more heterogeneous exposures, as otherwise acknowledged by Authors themselves.

The main text was amended as follows:

a) row 336 Conversely, the large prospective study from Torén et al. [47] was excluded from pooled analyses as the authors considered the exposure to metal fumes and welding tasks within a broader range of exposures associated with the construction industry

b) row 350 Again, the large study from Torén et al. [9] was not included in the pooled estimates because of the exposure assessment, based on job exposure matrix and not consistent with the other reports [8,13,34]. 

c) row 557 

Finally, our estimates on the mortality associated with pneumococcal pneumonia were affected by the inconsistence of the high-quality study from Torén et al. [9, 47] with other reports when dealing with the exposures and occupational settings. While other studies reported their estimates on professional welders [29,33,35], the report from Torén et al. [47] included a total of 8 deaths associated with pneumococcal pneumonia (RR 5.77, 95%CI 1.53 to 21.73) which occurred in construction industries. In other words, the sample included a pool of workers exposed to a broad range of occupational respiratory risk factors, with extensive overlap with inorganic dusts, chemicals, and wood dust. Similarly, the study on IPD from Torén et al. [9] based on Swedish registries reported ad increased occurrence of this condition among workers exposed to metal fumes (OR 2.24, 95%CI 1.41 to 3.35 calculated by means of logistic regression analysis), but again it should be stressed that such estimates were not specifically calculated on professional welders, rather on workers exposed to welding fumes in a broader range of occupational tasks.

Introduction Line 65: The authors should also include cochlear implants and liquor fistulas as risk factors for IPD

ANSWER: Done, thank you.

Table 3: The authors should define the abbreviations Conf. and Prob.

ANSWER: Thank you; we've amended Conf. --> Confirmed; Prob. --> Probable.

In the end, we thank you for your accurate contribution to our study, and we're confident about the eventual acceptance of our systematic review for an upcoming pubblication on Vaccines.

Round 2

Reviewer 2 Report

In the revised version of the manuscript the authors addressed all comments of my first review. I only recommend to use German "Ständige Impfkommission" (not Ständigen Impfkommission) in the abstract and the manuscript.

I only recommend to use German "Ständige Impfkommission" (not Ständigen Impfkommission) in the abstract and the manuscript.